# Association between Falls, Fear of Falling and Depressive Symptoms in Community-Dwelling Older Adults

**DOI:** 10.3390/healthcare12161638

**Published:** 2024-08-16

**Authors:** Maria Eduarda Armando Pereira, Gerson de Souza Santos, Clara Rabite de Almeida, Kethlyn Cristina Santos Nunes, Monalisa Claudia Maria da Silva, Helena José, Luís Sousa, Luciano Magalhães Vitorino

**Affiliations:** 1Faculty of Medicine of Itajubá, Itajubá 37502-138, MG, Brazil; mariaearmandop@gmail.com; 2Department of Medicine, Centro Universitário Ages, Paripiranga 48430-000, BA, Brazil; gerson.s.santos@ages.edu.br; 3Department of Nursing, Federal University of Juiz de Fora, Juiz de Fora 36036-900, MG, Brazil; clararabite95@gmail.com (C.R.d.A.); nunes.kethlyn@estudante.ufjf.br (K.C.S.N.); monalisa.silva@ufjf.br (M.C.M.d.S.); 4Atlântica School of Health, 2730-036 Barcarena, Portugal; hjose@uatlantica.pt (H.J.); luismmsousa@gmail.com (L.S.); 5Health Sciences Research Unit: Nursing (UICISA: E), Nursing School of Coimbra, 3004-011 Coimbra, Portugal; 6Comprehensive Health Research Centre, University of Évora, 7000-801 Évora, Portugal

**Keywords:** aged, older people, fear of falling, falls, depression

## Abstract

Background: Longevity increases pose public health challenges, especially in managing falls and their psychological impacts on older adults. Limited evidence exists on the relationship between a fear of falling (FOF), previous falls, and depressive symptoms among community-dwelling older adults. Objective: To evaluate the association between falls, FOF, and depressive symptoms in community-dwelling older adults. Methods: This cross-sectional study, conducted in 2018, included 400 older adults from a Basic Health Unit in São Paulo, Brazil. The Geriatric Depression Scale (GDS-15) and the International Falls Efficacy Scale (FES-I) were used, along with self-report questionnaires on fall history. Linear and logistic regression were used to analyze the relationships between variables. Results: The mean age was 75.2 (SD = 8.53) years, with 63.2% being female. Depressive symptoms were observed in 18.3% of the participants, while 90.5% reported a fear of falling (FOF). More than half (63.0%) experienced falls, with 49.5% occurring in the last year. Factors such as the female gender, negative health perceptions, and functional dependence were associated with depressive symptoms. Adjusted analyses indicated that both a fear of falling (FOF) (B = 0.043; *p* = 0.012) and a history of falls (B = 0.725; *p* = 0.015) were associated with depressive symptoms. Conclusions: Falls, FOF, and depressive symptoms are interlinked among older adults, underscoring the need for targeted interventions to improve their mental and physical health.

## 1. Introduction

The rapid increase in the older population poses significant challenges for healthcare professionals, researchers, and health systems worldwide. A major challenge is preserving the autonomy and quality of life (QoL) of this population, which is affected by the high incidence of chronic non-communicable diseases (CNCDs) [1]. Mental illnesses are a major type of chronic non-communicable disease (CNCD) that affect the quality of life (QoL) of older adults [2]. Frailty significantly elevates the risk of depression among older adults [3]. Additionally, falls represent a critical risk factor for mental health deterioration, as they are the leading cause of disability in this population [4,5]. A recent meta-analysis with 14 studies identified a strong relationship between falls (whether single or recurrent), or a fear of falling, and a higher risk of developing depression among older persons [6].

Depression is a debilitating condition that contributes to increased morbidity, mortality, and healthcare costs [7]. It affects a significant portion of the older population globally, with a prevalence of 35.1% worldwide and 40.8% in developing countries like Brazil. In Brazil, the average estimated prevalence of depressive symptoms among community-dwelling older persons is 21.0% [8,9]. A large European study with 67,603 participants, averaging 67.9 years in age, identified social isolation and a poorer self-assessment of one’s health status as the two strongest predictors of depression. Additionally, limitations in instrumental activities of daily living were significant predictors of depression in men, while the family burden was significant for women. Furthermore, mobility difficulties were found to increase the risk of falls and fear of falling for both sexes [10]. A meta-analysis of 17 studies involving 15,491 Brazilian older persons identified being female and having cardiovascular diseases as significant risk factors for depression [11].

Among the older population, falls are directly associated with the fear of falling and vice versa [12]. Generally, there are two ways to describe the fear of falling. The first is the definition focused on the fear itself, measuring the fearful anticipation of future falls. While an activity avoided out of fear can protect one from dangerous situations, if it exceeds a certain level, it becomes debilitating, starting a vicious cycle of inactivity, followed by the loss of physical conditioning and decreased confidence and social activity participation, ultimately enhancing the factors that increase the risk of depressive symptoms among older persons [13]. A Korean longitudinal study with 2933 older persons and 22 years of follow-up identified that both older persons who fell and did not have physical limitations and those who fell and had physical limitations experienced a substantial increase in the risk of depression [14].

Although the links between depressive symptoms and the risk of falls are not fully understood, several explanations exist. The use of selective serotonin reuptake inhibitors (SSRIs) and serotonin–norepinephrine reuptake inhibitors (SNRIs) significantly increases the risk of falls and fear of falling among older persons [15,16]. These medications can cause sedation, impaired balance, hyponatremia, cardiac arrhythmia, and drug-induced parkinsonism [16]. Depressive symptoms are associated with risk factors for falls, such as psychomotor slowness, leading to reduced walking speeds and imbalance [6]. People with depression may exhibit postural irregularities and changes in gait patterns, suggesting physiological causes for falls [17]. Additionally, depressive symptoms are linked to cognitive impairments that decrease attention, executive performance, and processing agility, increasing the risk of falls [18]. Geriatric depression is also associated with reduced appetite, weight loss, and muscle mass loss, further elevating the risk of falling [19].

Compared to other chronic non-communicable diseases (CNCDs), the prevalence of depressive symptoms in older persons, particularly when associated with factors such as a fear of falling, is relatively underexplored, especially in Brazil, as well as worldwide. The global prevalence of the fear of falling is 49.60%, with higher rates in developing countries (53.40%) than in developed countries (46.7%) and higher in patients (52.20%) than in community residents (48.40%) [20]. Falls are a significant public health issue among older adults, with an estimated 684,000 fatal falls occurring annually, making them the second leading cause of unintentional injury deaths globally (World Health Organization, 2021) [19]. A review of 104 studies with 36,740,590 participants found the global prevalence of falls in older people to be 26.5% (95% CI 23.4–29.8%) [21].

Effective fall prevention strategies, including education, safer environments, and policy implementation, are crucial [19]. In hospital and residential environments, falls are frequent but are challenging to prevent. An Italian study showed varying compliance with guidelines, indicating a need for standardized fall management and prevention strategies [20]. The lack of comprehensive evidence on the intersection of depression, a fear of falling, and falls in older adults may lead to inconsistent and suboptimal care. Depression is often underdiagnosed and undertreated in primary care settings, highlighting the necessity of addressing these gaps to improve healthcare outcomes. Gambaro et al. [6] support the association between a fear of falling and depressive symptoms, showing that various pieces of epidemiological evidence link depressive symptoms in older adults to numerous risk factors for falls, including psychomotor delays that can reduce walking speeds and impair balance. Understanding these connections is crucial in improving QoL and promoting better aging among older adults. 

Although there have been studies on this topic, to our knowledge, there are no studies focusing on community-dwelling older adults in Brazil using validated and reliable scales for the fear of falling and depressive symptoms. This would allow for meaningful comparisons and strengthen the practice of comprehensive geriatric assessment. The practical contributions of our study include offering valuable data that can inform healthcare policies and interventions aimed at improving the mental and physical health of older adults. Thus, this study aims to assess the association between falls, a fear of falling, and depressive symptoms among Brazilian community-dwelling older adults.

## 2. Materials and Methods

### 2.1. Study Design

This is a cross-sectional study that is part of a multidimensional study with older persons (aged 60 years or more) registered at a Basic Health Unit (BHU) in São Paulo, SP, Brazil. The study project was approved by the Research Ethics Committee of the Faculty of Medicine of Itajubá, (#2.468.315) on 17 January 2018. All participants signed an informed consent form.

### 2.2. Location and Sample

The study was conducted with older persons registered at the Marcus Belenzinho BHU located in the East Zone, the most populous area of São Paulo, SP. The sample size was calculated based on the following questions. (1) Would there be differences between the group of older persons who had a fear of falling and the group that did not have this fear regarding depressive symptoms? A meta-analysis showed a relative risk of 2.85 between a fear of falling and depression in the older population [6]. (2) Would there be differences between the group of older persons who had fallen in the last 12 months and the group that had not fallen in the last 12 months regarding depressive symptoms? A longitudinal study identified a risk of 2.73 between falling and depression [22]. The G*Power 3.1.9.7 program was used to calculate the statistical power of the analyses, with the aim of implementing multivariate linear and logistic regression models, with 3 independent variables (variables with *p* < 0.10 in bivariate analysis), with a relative risk of 2.85 between the fear of falling (considering the highest relative risk) and depressive symptoms, a two-tailed *p*-value, and 400 participants. The post hoc analysis showed the statistical power of the analyses to be 97.8%.

### 2.3. Data Collection and Inclusion and Exclusion Criteria

Data collection was conducted between February and August 2018 by a nurse during nursing consultations at the BHU, with an average duration of 40 minutes. Simple random sampling, based on the records of 5000 older persons registered at the BHU Marcus Belenzinho, was used to obtain the study participants. The inclusion criteria encompassed participants aged 60 years or older and registered at the BHU Belenzinho. Individuals with severe physical limitations (e.g., difficulty feeding themselves, bathing, or using the bathroom or severe respiratory difficulties) and a medical diagnosis of moderate to severe cognitive deficits or dementia were not included.

### 2.4. Measurement 

Depressive Symptoms: The abbreviated Geriatric Depression Scale with 15 items (GDS-15) was used to investigate the participants’ depressive symptoms. This scale was developed by Yesavage in 1983 [22] and validated in Portuguese in 2005 [23]. It is a fifteen-item scale, with two response options (yes; no). The score ranges from 0 to 15 and classifies patients without depressive symptoms (scores of 0 to 5) and with depressive symptoms (scores of 6 or greater) [22,23]. Example questions include “Are you satisfied with your life?” (yes/no), “Do you often feel helpless?” (yes/no), and “Do you feel your life is empty?” (yes/no). The sensitivity (86.5%) refers to the scale’s ability to correctly identify those with depressive symptoms, while the specificity (63.3%) refers to the scale’s ability to correctly identify those without depressive symptoms. In the Brazilian validation, the GDS-15 showed good reliability, with a Cronbach’s alpha of 0.81 [24].

### 2.5. Data Collected 

Fear of Falls: We used the Falls Efficacy Scale—International (FES-I), developed by Yardley in 2005 [25]. This scale was validated in 2010 for the Brazilian context [26]. The FES-I assesses the fear of falling in 16 different daily activities, including items such as moving into or out of a chair, walking in places with crowds, and cleaning the house (e.g., sweeping, vacuuming, dusting). The Falls Efficacy Scale—International (FES-I) scoring ranges from 16 (not concerned) to 64 (extremely concerned), with each item measured on a four-point Likert scale: 1 = not at all concerned, 2 = somewhat concerned, 3 = fairly concerned, and 4 = very concerned. The cutoff points for the fear of falling were as follows: 16–22 = low concern and 23–64 = high concern. In the Brazilian validation, the FES-I showed excellent reliability, with a Cronbach’s alpha equal to 0.96 [26].

Fall History (self-report). To investigate the fall history in the last 12 months, the following question was asked: “Have you fallen at home, on the street, or elsewhere in the last 12 months?” (yes or no).

Sociodemographic Variables. We included the sociodemographic and health variables of the elderly participants, with notable divisions in each category. These included age (60–69, 70–79, and ≥80 years), gender (female and male), income (less than and more than 1 minimum wage), health status (poor to excellent), the presence of chronic diseases (yes and no), the daily use of medication (yes and no), polypharmacy (yes and no), functional independence according to the Katz Index of Activities of Daily Living (dependent and independent), functional independence according to the Lawton Instrumental Activities of Daily Living Scale (dependent and independent), fall history (have fallen and have never fallen), fear of falling (afraid and not afraid), and depressive symptoms (yes and no). This detailed structure provided a comprehensive basis for the analysis of the relationships between depression, the fear of falling, and the characteristics of the participants.

Basic activities of daily living were evaluated using the Katz Index, created by Sidney Katz in 1976 and validated in Portuguese in 2008 [27]. It is an instrument that assesses the basic activities of daily living (ADL) according to the degree of independence in the outcomes of six ADL functions (bathing, dressing, going to the toilet, transferring, continence, and feeding). Example questions include “Can you bathe yourself without assistance?” (yes/no), “Can you dress yourself without help?” (yes/no), and “Can you use the toilet independently?” (yes/no). The score ranges from 0 to 6 points and classifies patients as independent (score zero) and dependent (score greater than or equal to 1). Its reliability and internal consistency as assessed by Cronbach’s alpha ranged from 0.80 to 0.92 [27].

Instrumental activities of daily living: Lawton Scale—devised by Lawton and Brody in 1969 and validated in Portuguese in 2008 [28]. It is used to assess the instrumental activities of daily living (IADL) according to scores ranging from 0 to 21. Example questions include “Can you prepare your own meals?” (yes/no), “Can you manage your own medications?” (yes/no), and “Can you handle your own finances?” (yes/no). It classifies patients as dependent (score less than or equal to 20) and independent (score equal to 21) [28]. The internal consistency measured according to Cronbach’s alpha ranged from 0.90 to 0.93 [28] and the reproducibility was considered stable and objective [29].

### 2.6. Statistical Analysis

Data were analyzed using SPSS version 26 (SPSS Inc., Chicago, IL, USA), with all analyses conducted by the research coordinator, L.M.V. Descriptive statistics included absolute and relative values for categorical variables and measures of central tendency for continuous variables on the GDS-15 and FES-I scales.

For inferential analysis, we employed both linear and logistic regression models. First, simple linear regression was used to examine the association between depressive symptoms (GDS-15) and the fear of falling or fall history. Next, multiple linear regression was used to adjust the depressive symptoms for significant sociodemographic and health variables (*p* < 0.05)—specifically, gender, self-perception of health, and ADL. 

The complete linear models employed were as follows.

Model 1: Depressive Symptoms = β0 + β1 × FES-I + ϵ.Model 2: Depressive Symptoms = β0 + β1 × FES-I + β2 × Gender + ϵ.Model 3: Depressive Symptoms = β0 + β1 × FES-I + β2 × Gender + β3 × Self-Perception of Health Status + β4 × ADL + ϵ.Model 4: Depressive Symptoms = β0 + β1 × History of Falling + ϵ.Model 5: Depressive Symptoms = β0 + β1 × History of Falling + β2 × Gender + ϵ.Model 6: Depressive Symptoms = β0 + β1 × History of Falling + β2 × Gender + β3 × Self-Perception of Health Status + β4 × ADL + ϵ.

Finally, logistic regression models were utilized to analyze depressive symptoms (coded as 0 = without or 1 = with depressive symptoms) with independent variables such as a fear of falling or fall history, adjusting for the same covariates as in the multiple linear regression.

These models assessed the independent effects of a fear of falling and fall history on depressive symptoms, while controlling for sociodemographic and clinical variables. A significance level of 5.0% was set for all tests, with 95.0% confidence intervals.

## 3. Results

Out of the total of 488 participants invited to take part in this study, 400 completed all items of the questionnaires, 50 participants did not meet the inclusion criteria, and 38 refused to participate in the study. The sociodemographic and health characteristics are shown in Table 1. The average age of the participants was 75.2 (SD = 8.53) years; the majority were between 70 and 79 years old (39.8%) and female (63.2%), with poor/regular self-perceptions of health (62.6%) and with chronic diseases (92.2%). Regarding medication use, 90.7% of the participants used medication daily; of these, 37.0% exhibited polypharmacy (≥5 medications/day). We identified that 27.0% of the participants were dependent on ADL and 39.5% on IADL. The majority of the older persons reported having suffered a fall (63.5%), with half of the falls occurring in the last year (49.5%). The majority of the older adults (90.5%) also exhibited a fear of falling (FES-I ≥ 23), while the prevalence of depressive symptoms (GDS ≥ 6) was 18.3%.

Table 2 shows the mean scores of the FES-I scale and GDS-15.

Table 3 presents the results of the simple linear regression. Being female (B = −0.954; *p* = 0.002), having a worse perception of one’s health (B = −0.555; *p* = 0.009), and being dependent for ADL (B = −0.893; *p* = 0.036) were associated with depressive symptoms.

The results of the adjusted linear regression are shown in Table 4. After adjusting for gender, the self-perception of one’s health status, ADL, higher levels of fear of falling (B = 0.043; *p* = 0.012), and a history of falling (B = 0.725; *p* = 0.015) were associated with depressive symptoms.

The adjusted and unadjusted logistic regression (Table 5) indicated that older persons with a higher level of fear of falling (OR = 1.03, *p* = 0.005) and a history of falls (OR = 1.64, *p* = 0.036) showed a greater propensity for depressive symptoms.

## 4. Discussion

This study explored how the fear of falling and a history of falls can impact mental health—specifically, depressive symptoms—in community-dwelling older persons. We identified that, after controlling for sociodemographic variables, an increase in the fear of falling and the presence of a history of falls were strongly associated with a higher prevalence of depressive symptoms. This finding is consistent with previous studies, such as a systematic review [8,23] and longitudinal research [24], reinforcing the relationship between these factors and depressive symptoms in older adults. In a meta-analysis that included 31 studies with 70,868 elderly people living in the community, the factors associated with falls were age, the female sex, a fear of falling, a history of falls, unclear vision, depression, and balance disorders [25]. A Greek study also found similar results and demonstrated that falling was associated with declining balance, depression, the female gender, and higher education [26]. However, a prospective study involving 6862 older people in Ireland found that falls increased the risk for incident symptoms of anxiety and depression [26].

We observed that a fear of falling among older adults is a prevalent reality that intensifies with age. This trend is influenced by both biological age and the subjective perception of one’s age [27]. Our results point to a significant association between the fear of falls and the development of depressive symptoms. This relationship suggests that depressive symptoms may emerge as a direct consequence of social isolation and the loss of independence, both resulting from the limitations imposed by the fear of falls [6,28]. In a study involving 3443 individuals, poorer social engagement and network contact were found to be associated with a higher likelihood of falls, while a poorer neighborhood context was associated with a higher likelihood of fall injuries. Furthermore, social engagement mediated a significant portion of the effect of depression on the occurrence of falls, and the neighborhood context mediated a portion of the effect of depression on fall injuries [29].

Conversely, depression itself can exacerbate the fear of falling by reducing older adults’ confidence in their physical abilities, creating a vicious cycle that affects both mental and physical health [6]. In a study conducted in Brazil, it was found that being a woman and having a higher number of self-reported morbidities, worse physical performance, and a higher number of depressive symptoms were associated with a higher level of fear of falling. In addition, it was found that physical inactivity, mediated by a higher number of morbidities, worse physical performance, and a higher number of depressive symptoms, was associated with a higher level of fear of falling [30]. Living arrangements, such as living alone, with family, or in an elderly home, can significantly influence these dynamics. For instance, living alone may increase the risk of social isolation and falls, while living with family can provide support but also create a potential family burden [12,31].

A scoping review covering 46 studies, supplemented by a Korean longitudinal study with 62,363 participants over 16 years, highlighted the complex interrelation between FOF and depressive symptoms in older adults [32,33]. The finding of an association between a fear of falling and depressive symptoms in 95% (19/20) of the studies analyzed in the scoping review underscores the association between a fear of falling and depressive symptoms. This phenomenon highlights that depression is not merely a psycho-emotional aspect of the fear of falling but also a factor that intensifies the associated risks, suggesting a vicious cycle where the fear of falling and depression are mutually intensified, exacerbating frailty and increasing the vulnerability of older adults. These findings underscore the urgent need for therapeutic approaches that encompass both mental health treatment and fall prevention strategies, aiming to mitigate this risk in the older population. Such approaches could include cognitive–behavioral therapy (CBT) to address both depression and the fear of falling, tailored physical exercise programs like Tai Chi and strength training to improve balance and confidence, and multifactorial interventions that combine medical, psychological, and physical therapy to meet the comprehensive needs of older adults. These interventions should be designed to reduce both the fear of falling and depressive symptoms, thereby improving the overall quality of life [31].

Our findings also reinforce the link between a history of falls and depressive symptoms in older adults, in line with previous studies. The systematic review by Xu et al. highlights depression as a significant risk factor for falls (RR 4.34) [34], surpassing other common conditions such as heart disease and hypertension. Factors related to depression, such as social isolation, apathy, physical inactivity, physical impairment, and the use of antidepressants, increase the risk of falls.

The relationship between depressive symptoms and falls, although well established, reveals complexities yet to be unraveled. Risk factors for falls, such as psychomotor delays, are associated with depressive symptoms, affecting individuals’ locomotion speeds and balance. Gambaro et al. [6] underline a growing interest in the interaction between depression, falls, and a fear of falling, evidencing a bidirectional relationship: falls negatively affect independence and well-being, while depression can lead to psychomotor delays and changes in posture and gait. A longitudinal study in Korea showed a significant correlation between depressive symptoms and the number of previous falls, with depression remaining a risk factor even after adjusting for other variables, including the fear of falling [32].

### 4.1. Study Limitations and Strengths

While addressing the relationship between a fear of falling, a history of falls, and depressive symptoms in older adults, this study provides significant findings but also presents some important limitations that should be considered. Firstly, the cross-sectional nature of the study limits the ability to establish causal relationships between the variables. Additionally, factors such as financial issues, the family history, the home environmental conditions, and pre-existing diseases were not associated with the outcomes but could offer a deeper understanding of the dynamics at play. Another limitation is the participant group, which, although extensive, may not fully represent the diversity of the Brazilian older population, particularly in different cultural and socioeconomic contexts. This introduces a potential selection bias, as the sample was drawn from older persons registered at a BHU in São Paulo, Brazil and may not be representative of the Brazilian population of older adults. Future studies should aim to include a more representative sample to enhance the external validity of the findings.

Furthermore, the low adjusted R^2^ values in our regression models indicate that, while the predictors are statistically significant, they explain only a small portion of the variance in depressive symptoms. This suggests that other factors, not included in our models, also contribute to depressive symptoms among older adults. The complexity of depressive symptoms, which are influenced by a wide range of biological, psychological, and social factors, could account for the relatively low explanatory power of our models.

However, the study also has strengths that should be highlighted. It includes a considerable sample of older adults, of both sexes and with varied socioeconomic conditions, enriching the analysis and allowing for a more comprehensive understanding of the factors associated with depressive symptoms in older adults. Moreover, the use of validated instruments, such as the Geriatric Depression Scale (GDS-15) and the Falls Efficacy Scale—International (FES-I), lends methodological robustness to the findings.

### 4.2. Implications for Clinical Practice, Policy, and Education

The results of this study are significant for healthcare professionals, especially in geriatrics, gerontology, and primary healthcare. With the increasing older population, managing the fear of falling and preventing falls to reduce depressive symptoms and improve the well-being of older adults is vital. The research emphasizes the importance of comprehensive geriatric assessments that consider the fear of falling, fall prevention, and depressive symptoms as key elements of older adult health. 

For professionals in primary care, the findings highlight the need for preventive approaches. The risk of falling can be reduced by implementing exercise programs focused on balance and strength training, combining interventions targeting modifiable risk factors [35], as well as interventions to mitigate the fear of falling [36]. In a review of fall risk assessment instruments, 38 instruments were identified, with 23 tools targeting hospitalized patients, eight aimed at risk assessment in home residents, and seven that can be applied in both contexts. In this review, the FES-I is recommended for the assessment of the fear of falling. However, as the nature of the fall risk is multifactorial, there is no “ideal” instrument in this sense. It is recommended to apply a combination of instruments, associated with direct and in-depth analysis carried out by the health professional [37].

Taking into account the above, implementing screening programs and interventions to identify and treat the fear of falls early is crucial to avoid negative consequences such as social isolation and a loss of independence. A fear of falling can significantly influence physical function in older adults and consequently increase the risk of future falls. Therefore, healthcare professionals should assess the fear of falling and consider implementing strategies to reduce the fear of falling as part of a comprehensive, person-centered care plan [38]. These programs must have psycho-educational interventions that allow the empowerment and training of the older adult and their family caregiver/informal caregiver, in self-care and activities of daily living, as well as in physical exercise (Tai Chi, yoga, balance training, or strength and resistance training) and fall prevention [39,40,41,42].

Integrating these assessments into regular care would help to promote the health of older adults and prevent mental and physical health problems. These interventions regarding the prevention of the risk of falling and fear of falling must be based on the policies of the World Health Organization, which aim to contribute to healthy aging, in order to prevent diseases, promote physical and mental health, and maintain functional capacity [43].

These results also have implications for education policies. In this sense, modules on assessment and intervention aimed at preventing the risk of falls, fear of falling, and depressive symptoms in community-dwelling older adults should be integrated into undergraduate and postgraduate courses, in lectures or modules on physical exercise prescription and physical activity in elderly people. With better-prepared professionals, this will contribute to safer and more person-centered care.

## 5. Conclusions

A fear of falling, a history of falls, and depressive symptoms are complex and interconnected factors that have a significant impact on the lives of older people. Previous research indicates that a fear of falling is often associated with previous falls and can lead to adverse consequences such as restricted physical activity and a loss of functional independence, and the experience of falls is often followed by physical and emotional complications, with potential sequelae ranging from physical injuries to the development of mental disorders such as depression. 

Our findings establish a significant association between the fear of falling, a history of falls, and depressive symptoms among older people, highlighting their high prevalence and negative impact on mental health. These results underscore the importance of developing integrated strategies focused on fall prevention, managing the fear of falling, and treating depression, aimed at enhancing the physical and mental health of this population. It is undeniable that the relationship between falls, a fear of falling, and depressive symptoms in older people is a crucial area of research, requiring holistic and personalized interventions to improve the health and well-being of this vulnerable population.

## Figures and Tables

**Table 1 healthcare-12-01638-t001:** Characterization of participants (N = 400).

Variable	n	%
Age		
60–69	104	26.0
70–79	159	39.8
≥80	137	34.2
Gender		
Female	253	63.2
Male	147	36.8
Family income		
<1 salary	51	12.8
>1 salary	349	87.2
Health status		
Awful	89	22.3
Regular	161	40.3
Good	120	30.0
Excellent	30	7.5
Chronic illness		
Yes	369	92.2
No	31	7.8
Medication use		
Yes	363	90.7
No	37	9.3
Polypharmacy		
Yes	148	37.0
No	252	63.0
BADL ^1^ (Katz)		
Dependent	108	27.0
Independent	292	73.0
IADL ^2^ (Lawton)		
Dependent	158	39.5
Indepentend	242	60.5
History of falls ^3^		
Fell	252	63.0
Never fell	148	37.0
Last fall		
<1 year	198	49.5
>1 year	51	12.8
Never fell	151	37.8
Fear of falling (FES-l ^4^)		
With fear (FES-I: 23–64)	362	90.5
Without fear (FES-I: 16–22)	38	9.5
Depressive symptoms		
Yes (GDS ≥ 6)	73	18.3
No (GDS ≤ 5)	327	81.8

^1^ Basic Activities of Daily Living; ^2^ Instrumental Activities of Daily Living; ^3^ Last Year; ^4^ Falls Efficacy Scale—International.

**Table 2 healthcare-12-01638-t002:** Fear of falling and depressive symptom scores of study participants (N = 400).

Variable	Mean (SD)	95% CI
Depressive Symptoms (GDS-15 ^1^)	5.98 (2.89)	5.72–6.28
Fear of Falling (FES-I ^2^)	41.84 (12.99)	40.65–43.10

(GDS ≥ 6) ^1^ Geriatric Depression Scale-15; ^2^ Falls Efficacy Scale—International.

**Table 3 healthcare-12-01638-t003:** Associations between the independent variables and depressive symptoms among older participants (N = 400).

Variable	B (SE)	Beta	*p*-Value
Age (60–69)	−0.018 (0.023)	−0.054	0.427
Gender (female)	−0.954 (0.303)	−0.159	0.002
Family income (<1 minimum wage)	−0.105 (0.242)	−0.024	0.664
Health status (poor)	−0.555 (0.213)	−0.138	0.009
Chronic disease (yes)	−0.633 (1.002)	−0.059	0.528
Use of medications (yes)	−0.037 (0.927)	−0.004	0.968
Polypharmacy (yes)	0.247 (0.305)	−0.041	0.419
ADL ^1^ (dependent)	−0.893 (0.425)	−0.137	0.036
IADL ^2^ (dependent)	−0.170 (0.363)	−0.029	0.639

^1^ Basic Activities of Daily Living; ^2^ Instrumental Activities of Daily Living.

**Table 4 healthcare-12-01638-t004:** Associations between fear of falling and depressive symptoms adjusted for sociodemographic variables (N = 400).

Variable	B (SE)	Beta	*p*-Value	Adjusted R^2^
FES-I ^1^ (average)	0.053 (0.011)	0.230	0.001	0.013
FES-I ^1^ (sociodemographic)	0.057 (0.016)	0.244	0.001	0.018
FES-I ^1^ (sociodemographic + clinical)	0.043 (0.17)	0.184	0.012	0.095
History of falling (yes)	1.017 (0.295)	0.170	0.001	0.027
History of falling (sociodemographic *)	0.853 (0.335)	0.143	0.011	0.130
History of falling (sociodemographic * + clinical)	0.725 (0.361)	0.188	0.015	0.150

^1^ Falls Efficacy Scale—International. * Adjusted for sociodemographic (gender) and clinical (self-perception of health status and basic activities of daily living) variables.

**Table 5 healthcare-12-01638-t005:** Unadjusted and adjusted logistic regression analyses between independent variables and depressive symptoms (N = 400).

Variable	OR (CI95%)	*p*-Value
FES-I ^1^ (average)	1.04 (1.022–1.05)	<0.001
FES-I ^1^ (sociodemographic)	1.03 (1.01–1.05)	0.001
FES-I ^1^ (sociodemographic + clinical)	1.03 (1.01–1.05)	0.005
History of falling (yes)	2.04 (1.34–3.09)	0.001
History of falling (sociodemographic *)	1.72 (1.11–2.67)	0.015
History of falling (sociodemographic * + clinical)	1.64 (1.03–2.62)	0.036

^1^ Falls Efficacy Scale—International. * Adjusted for sociodemographic (gender) and clinical (self-perception of health status and basic activities of daily living) variables. GDS ≥ 6: Yes = 1; No = 0.

## Data Availability

Data will be available upon request (lucianoenf@yahoo.com.br).

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
