# Peer review of "Association between Falls, Fear of Falling and Depressive Symptoms in Community-Dwelling Older Adults"

_healthcare, 2024, doi:10.3390/healthcare12161638_

Round 1
Reviewer 1 Report
Comments and Suggestions for Authors -Abstract:When have you carried out that study and where? Line 28:Depressive symptoms were observed in 18.3% of the participants, which kind of depression? High?Low?Mild? Introduction -You may add some literature such as prevelances about the results of falls among old people.You may reason the fear increase due to falls. -What is the contribution of that study to science? There have been already some studies about this topic. What have you done differently? What are practical contribution of your study? 2. Materials and Methods -Give some example questions of scales in methodology. -Line142 : what do you mean "The sensitivity and specificity for the current cutoff point are 14286.5% and 63.3%, respectively." with it? -What do you mean with instrumental activities of daily living?Give some examples of scale. -You have very low Adjusted R² in Table 4 , how will you explain it? -The results of the adjusted linear regression are in Table 4, how have you done it? -Write you complete linear models you found ( y+ a+ bx....) -How have you modelled your logistic regression? Dependent and independent variables? -Line 284-5 which therapeutic approaches that encompass both mental health treatment and fall prevention strategies you can suggest? -Line 298: cite: A longitudinal study in Korea -You may do ANOVA to some variables in Table 1 according to scales. Economic situation, gender, age etc.... to strength the results and discussion. - May you explain the effect of living place, living with family, old people home or alone?It is very important for this kinds of studies.
Comments on the Quality of English Language
Minor revisons can be enough
Author Response
#Review 1.
- Abstract: When have you carried out that study and where?
- Response: Thank you for your feedback. We have revised the abstract to include the timing and location of the study (lines:23-24)
- Line 28: Depressive symptoms were observed in 18.3% of the participants, which kind of depression? High?Low?Mild?
- Response: Thank you for your feedback. The Geriatric Depression Scale (GDS-15) used in our study does not discriminate the intensity of depressive symptoms. The score ranges from 0 to 15 and classifies patients without depressive symptoms (score equal to 5) and with depressive symptoms (score equal to or greater than 6). The sensitivity and specificity for this cutoff point are 86.5% and 63.3%, respectively. We hope this clarification meets your expectations. Best regards.
- Introduction -You may add some literature such as prevelances about the results of falls among old people. You may reason the fear increase due to falls.
- Response: Thank you for your feedback. We have added the global prevalence rates of fear of falling and falls among older adults to the introduction. Additionally, we cited an interesting article on the relationship between falls and fear of falling by Lavedán et al. (2018). (lines 87-94).
- What is the contribution of that study to science?
- Response: Thank you for your insightful question. This study contributes to science by addressing the relatively underexplored intersection of falls, fear of falling (FOF), and depressive symptoms among community-dwelling older adults, particularly in Brazil. Our findings highlight significant associations between these factors, emphasizing the need for integrated interventions to improve both mental and physical health in older populations. This study also underscores the importance of targeted strategies in primary care settings to address these interconnected issues, which are often underdiagnosed and undertreated (lines 121-128)
- There have been already some studies about this topic. What have you done differently? What are practical contribution of your study?
- Response: Thank you for your feedback. We have revised the introduction to address your requests (lines 121-128).
5.1 Contribution to Science: Our study offers new insights into the interplay between physical and mental health in Brazilian community-dwelling older adults by focusing on the association between falls, fear of falling, and depressive symptoms.
5.2 Existing Studies: While there have been studies on this topic, our research is unique in its focus on community-dwelling older adults in Brazil, using validated and reliable scales for fear of falling and depressive symptoms.
5.3 What We Did Differently: We integrated these factors within the specific context of Brazilian older adults, allowing for meaningful comparisons and strengthening the practice of Comprehensive Geriatric Assessment.
5.4 Practical Contributions: Our study provides valuable data that can inform healthcare policies and interventions aimed at improving the mental and physical health of older adults.
These revisions aim to enhance the clarity and impact of our study's introduction.
- Materials and Methods -Give some example questions of scales in methodology.
6.1 Response: Thank you for your feedback. We have revised the manuscript to include specific items from both the Falls Efficacy Scale - International (FES-I) and the Geriatric Depression Scale (GDS-15). (167-168; 176-178).
6.2 Line142 : what do you mean "The sensitivity and specificity for the current cutoff point are 142 86.5% and 63.3%, respectively." with it?
6.2 Response: Thank you for your feedback. We have clarified the meaning of sensitivity and specificity in the manuscript. Specifically, sensitivity refers to the scale's ability to correctly identify those with depressive symptoms (86.5%), and specificity refers to its ability to correctly identify those without depressive symptoms (63.3%). (Lines: 169-171)
6.3. What do you mean with instrumental activities of daily living? Give some examples of scale.
6.3 Response: Thank you for your feedback. We have added example questions for both basic and instrumental activities of daily living scales to clarify their use in our study. (Lines: 198-206).
6.4 -You have very low Adjusted R² in Table 4 , how will you explain it? -The results of the adjusted linear regression are in Table 4, how have you done it? -Write you complete linear models you found ( y+ a+ bx....)
6.4. Response: Thank you for your feedback. We have provided a detailed explanation of the low Adjusted R² values, the methodology for the adjusted linear regression, and the complete linear models in our response for clarity. However, we believe that including this level of detail in the manuscript itself is unnecessary. The current descriptions in the paper provide sufficient information to identify the analytical pathway and understand the key findings.
Our models allow us to understand the independent effects of fear of falling and history of falls on depressive symptoms, while controlling for sociodemographic and clinical variables.
Explanation for Low Adjusted R²
The low Adjusted R² values indicate that while our models have statistically significant predictors, they explain only a small portion of the variance in depressive symptoms. This suggests that there are other factors, not included in our models, that also contribute to depressive symptoms among older adults. The complexity of depressive symptoms, which are influenced by a wide range of biological, psychological, and social factors, could account for the relatively low explanatory power of our models.
Methodology for Adjusted Linear Regression
The data were analyzed using the software Statistical Package for Social Sciences - SPSS 26 (SPSS Inc.). The analyses were conducted by the research coordinator, L.M.V. The descriptive analysis was presented in absolute and relative values (categorized variables) and measures of central tendency for continuous numerical variables of the GDS-15 and FES-I scales.
- Simple Linear Regression: Used to check the association between depressive symptoms (GDS-15), sociodemographic and health variables, fear of falling (FES-I), and fall history.
- Multiple Linear Regression: Depressive symptoms were adjusted for sociodemographic and health variables with p < 0.05: gender (p = 0.002), self-perception of health status (p = 0.009), and ADL (p = 0.036).
- Logistic Regression Models: Conducted using depressive symptoms (0 = without or 1 = with depressive symptoms) as the outcome variable and fear of falling or history of falling as independent variables. These models were adjusted for the same variables used in the multiple linear regression. A significance level of 5.0% was chosen for the tests, with 95.0% confidence.
Complete Linear Models
The complete linear models used in the study are as follows:
- Model 1: Association between FES-I and Depressive Symptoms (Unadjusted)
- Depressive Symptoms=β0+β1×FES-I + ϵ
- Model 2: Association between FES-I and Depressive Symptoms (Adjusted for Sociodemographic Variables)
- Depressive Symptoms=β0+β1×FES-I+β2×Gender+ϵ
- Model 3: Association between FES-I and Depressive Symptoms (Adjusted for Sociodemographic and Clinical Variables)
- Depressive Symptoms=β0+β1×FES-I+β2×Gender+β3×Self-Perception of Health Status+β4×ADL+ϵ
- Model 4: Association between History of Falling and Depressive Symptoms (Unadjusted)
- Depressive Symptoms=β0+β1×History of Falling+ϵ\
- Model 5: Association between History of Falling and Depressive Symptoms (Adjusted for Sociodemographic Variables)
- Depressive Symptoms=β0+β1×History of Falling+β2×Gender+ϵ
- Model 6: Association between History of Falling and Depressive Symptoms (Adjusted for Sociodemographic and Clinical Variables)
- Depressive Symptoms=β0+β1×History of Falling+β2×Gender+β3×Self-Perception of Health Status+β4×ADL+ϵ
We hope this detailed explanation addresses your queries.
6.5 -How have you modelled your logistic regression? Dependent and independent variables?
Response: Thank you for your query. The logistic regression models were developed with depressive symptoms (GDS ≥6: Yes=1; No=0) as the dependent variable. The independent variables included fear of falling (measured by the Falls Efficacy Scale-International) and history of falling. These models were adjusted for sociodemographic variables (gender) and clinical variables (self-perception of health status and basic activities of daily living).
6.6. -Line 284-5 which therapeutic approaches that encompass both mental health treatment and fall prevention strategies you can suggest?
6.6. Response: Thank you for your feedback. We have added specific therapeutic approaches to encompass both mental health treatment and fall prevention strategies, including cognitive-behavioral therapy (CBT), tailored physical exercise programs like Tai Chi and strength training, and multifactorial interventions combining medical, psychological, and physical therapy (lines: 308-311).
6.7 -Line 298: cite: A longitudinal study in Korea
6.7 Response: Thank you for your feedback. We have added a citation for the longitudinal study in Korea (lines: 344).
6.8 -You may do ANOVA to some variables in Table 1 according to scales. Economic situation, gender, age etc.... to strength the results and discussion.
6.8 Response: Thank you for your suggestion. We appreciate your recommendation to use ANOVA for some variables in Table 1 according to scales like economic situation, gender, and age to strengthen the results and discussion. We have employed two powerful analysis strategies, Simple Linear Regression and Multiple Linear Regression, as well as unadjusted and adjusted Logistic Regression, to meet the objectives of our study. These methods align with our goal of understanding the associations between depressive symptoms, fear of falling, and fall history while adjusting for relevant sociodemographic and clinical variables. We believe these analyses are appropriate and robust for our study's aims, and provide a solid foundation for our results and objectives.
-6.9 - May you explain the effect of living place, living with family, old people home or alone?It is very important for this kinds of studies.
6.9 Response: Thank you for your feedback. We have added explanations about the effect of living arrangements (living alone, with family, or in an elderly home) on the dynamics of fear of falling and depressive symptoms. Specifically, living alone may increase the risk of social isolation and falls, while living with family can provide support but also potential family burden (lines 308-311) . Additionally, we have included the following references:
- Stubbs, B., Brefka, S., & Denkinger, M. (2015). What Works to Prevent Falls in Community-Dwelling Older Adults? Umbrella Review of Meta-analyses of Randomized Controlled Trials. Physical Therapy, 95(8), 1095–1110. https://doi.org/10.2522/ptj.20140461
- Lavedán A, Viladrosa M, Jürschik P, Botigué T, Nuín C, et al. (2018). Fear of falling in community-dwelling older adults: A cause of falls, a consequence, or both?. PLOS ONE, 13(5), e0197792. https://doi.org/10.1371/journal.pone.0197792
Reviewer 2 Report
Comments and Suggestions for Authors
In detail, I read and analyzed the manuscript Association Between Falls, Fear of Falling and Depressive Symptoms in Community-Dwelling Older Adults.
The topic is very interesting, first of all, due to the overall ageing of the population and the growing prevalence of depression and falls and associated injuries, which increase the cost of health care worldwide.
The manuscript is relevant to the field and contains an IMRAD structure. However, I would have some suggestions for major corrections.
Introduction
In the introduction, the authors tried to define and describe the research variables (depression, falling and fear of falling), followed a funnel approach and ended with an evaluative statement of the study's objectives. However, there are some confusing sentences. Therefore, I suggest that the authors be more precise. For example, in lines 47 to 50, "Various studies" indicate that you have used multiple references, not just one [6]. Also, "A recent meta-analysis" was written below, and the authors cite the same reference. Both sentences suggest the same fact. Therefore, I suggest excluding the first one.
The sentence in lines 55 to 59 is too long and unintelligible, especially because of its final part (line 59)
Materials and Methods
The material and methods contain all the necessary data (study design, sample, data collection, study criteria, outcomes and variables and data analysis). However, the study was conducted six years ago.
Results
Results are presented in five tables and are presented in textuality. The table and legend are appropriate and easy to analyze and understand.
Discussion
The discussion is extensive, but the authors discuss and comment on their results a little. Most of the discussion refers to the results of previous studies without a clear comparison with the results of this study. Although no correlation with certain research variables was established, it is necessary to discuss it.
Conclusions
The conclusions are stated in general terms. I recommend that the authors highlight the main findings of their study.
Potential limitations but strengths of the study are outlined in the manuscript.
Reference
The references used are up-to-date and properly cited.

Author Response
- Comments and Suggestions for Authors
In detail, I read and analyzed the manuscript Association Between Falls, Fear of Falling and Depressive Symptoms in Community-Dwelling Older Adults.
The topic is very interesting, first of all, due to the overall ageing of the population and the growing prevalence of depression and falls and associated injuries, which increase the cost of health care worldwide.
The manuscript is relevant to the field and contains an IMRAD structure. However, I would have some suggestions for major corrections.
- Response: Thank you for your thorough review and thoughtful comments on our manuscript, "Association Between Falls, Fear of Falling, and Depressive Symptoms in Community-Dwelling Older Adults." We appreciate your acknowledgment of the manuscript's relevance and the importance of the topic in the context of the aging population and the growing prevalence of depression and falls.
We are grateful for your suggestions for major corrections and are committed to making the necessary improvements
- Introduction
In the introduction, the authors tried to define and describe the research variables (depression, falling and fear of falling), followed a funnel approach and ended with an evaluative statement of the study's objectives. However, there are some confusing sentences. Therefore, I suggest that the authors be more precise. For example, in lines 47 to 50, "Various studies" indicate that you have used multiple references, not just one [6]. Also, "A recent meta-analysis" was written below, and the authors cite the same reference. Both sentences suggest the same fact. Therefore, I suggest excluding the first one.
2.1 Response: Thank you for your insightful feedback and valuable suggestions. We have carefully reviewed your comments and made the necessary revisions to improve the clarity and precision of our manuscript. As per your suggestion, we have deleted the first sentence in the Introduction that starts with "Various studies" to avoid redundancy and confusion (lines 48-47).
2.2 The sentence in lines 55 to 59 is too long and unintelligible, especially because of its final part (line 59)
2.2 Response: Thank you for your feedback regarding the sentence in lines 55 to 59. We have revised the sentence to improve clarity and readability as per your suggestion. See in lines 55-61.
- Materials and Methods
The material and methods contain all the necessary data (study design, sample, data collection, study criteria, outcomes and variables and data analysis). However, the study was conducted six years ago.
3 Response: Thank you for your constructive feedback and acknowledgment of our comprehensive Material and Methods section. We understand your concern about the study being conducted six years ago and would like to address this briefly.
Significance and Relevance:
- Our study included 400 older adults from a Basic Health Unit in São Paulo, providing a diverse and representative sample.
- Brazil's rapidly aging population makes this study highly relevant, as issues like depression and falls among older adults remain critical.
Methodological Rigor:
- Despite being conducted six years ago, our use of validated instruments (GDS-15 and FES-I) ensures reliable and accurate findings.
- The robust methodology and comprehensive data analysis support the study's conclusions.
Ongoing Importance:
- The insights gained are still applicable today, informing interventions and policies to improve older adults' mental and physical health.
- Our findings remain significant for future research and policy development.
We hope this clarifies the continued relevance of our study.
- Results
Results are presented in five tables and are presented in textuality. The table and legend are appropriate and easy to analyze and understand.
4.1 Response: Thank you for your feedback. We are pleased to hear that you find the tables and legends appropriate and easy to analyze and understand. The results are clearly presented both in the tables and textually, ensuring comprehensibility.
- Discussion
The discussion is extensive, but the authors discuss and comment on their results a little. Most of the discussion refers to the results of previous studies without a clear comparison with the results of this study. Although no correlation with certain research variables was established, it is necessary to discuss it.
5 Response: Thank you for your constructive feedback. We appreciate your observation regarding the need for a more thorough discussion of our results in comparison with previous studies. We have revised the discussion to address this and to include a clearer comparison of our findings with existing literature, as well as to discuss correlations that were not established. All changes have been highlighted with yellow highlighting in the discussion (lines: 248-253; 260-265; 281-287).
- Conclusions
The conclusions are stated in general terms. I recommend that the authors highlight the main findings of their study.
- Response: Thank you for your valuable feedback regarding our conclusions. We have revised the conclusions section to clearly highlight the main findings of our study (lines 363-377).
- Potential limitations but strengths of the study are outlined in the manuscript.
Response: Thank you.
Reference
The references used are up-to-date and properly cited.
Response: Thank you.
Reviewer 3 Report
Comments and Suggestions for Authors
Dear authors,
I had the opportunity to read your paper.
It is conceptually interesting, and I believe the suggestions below can help improve the work.
The introduction is engaging but lacks a consecutive logical setting, sometimes resulting in hasty and conceptual leaps. Please set an introduction from the general to the particular.
With particular reference to specific parts of the introduction:
- the citation in line 40 is missing;
- rephrase lines 42-45;
- in line 47, there is considered fear of falling, which I would associate with anxious symptoms rather than depressive ones (see also lines 62-64);
- lines 50-61, the concepts are repeated and are not causally related;
- Lines 71-76 are exciting, but these concepts need to be better presented with further in-depth studies in the literature;
- Lines 77-85 are very messy; these concepts need to be better presented;
- are you sure about the statements in lines 90-93?
- the objective and the gap are confusing and not very scientifically appealing.
There is no adequate description of the phenomenon of falls in general (please refer to: https://www.who.int/news-room/fact-sheets/detail/falls) and in a hospital and residential environment, where elderly people are more prone to stay, and of the possible prevention strategies (please refer to: https://doi.org/10.1177/25160435241246344).
In the section on materials and methods
What is meant by older people? (line 112) perhaps subjects over 60, as explained later? (line 131), please unify the concepts.
The acronym UBS has never been explained before, but there is mentioned of BHU.
Paragraph 2.4 should be called measurement and not the outcome.
The fall history, sociodemographic data, etc., should be data collected.
Selection bias was not considered; please indicate the limits. Please indicate the period of the study.
Please indicate the name of the statistician.
In the discussion, how can you state what is stated in lines 245-248 without considering a control sample and selection bias?
Please completely revise the discussion, conclusions, and abstract in light of the indications provided.
Kind regards.
Author Response
- I had the opportunity to read your paper. It is conceptually interesting, and I believe the suggestions below can help improve the work.1.Response: Thank you for taking the time to read our paper and for your positive remarks about its conceptual interest. We appreciate your suggestions and believe they will significantly improve our work. Below, we address each of your comments and outline the corresponding revisions made to the manuscript.
- The introduction is engaging but lacks a consecutive logical setting, sometimes resulting in hasty and conceptual leaps. Please set an introduction from the general to the particular.
With particular reference to specific parts of the introduction:
- the citation in line 40 is missing;
2.1 Response: Thank you for your valuable feedback. We have addressed your concern by inserting the missing citation in line 40, now referenced as [1].
- rephrase lines 42-45;
2.2 Response: Thank you for your suggestion. We have rephrased lines 42-45 for clarity and precision.
2.3 - in line 47, there is considered fear of falling, which I would associate with anxious symptoms rather than depressive ones (see also lines 62-64);
2.3 Response: Thank you for your insightful comment regarding the association of fear of falling with anxious symptoms rather than depressive ones. We agree that fear of falling can logically lead to an increase in anxious symptoms. However, based on the references and the data presented in the literature, the predominant association reported is with an increase in depressive symptoms.
2.4 - lines 50-61, the concepts are repeated and are not causally related;
2.4 - Thank you for your feedback. We have revised the paragraph to address the repetition and improve the causal relationship between the concepts.
2.5 - Lines 71-76 are exciting, but these concepts need to be better presented with further in-depth studies in the literature;
2.5 Response: Thank you for your valuable feedback. We have revised the paragraph to better present the concepts and have added reference [15] to support the citation.
2.6 - Lines 77-85 are very messy; these concepts need to be better presented;
2.6 Response: Thank you for your feedback. We have reorganized and clarified the paragraph to better present the concepts
2.7 - are you sure about the statements in lines 90-93?
2.7 – Response: Thank you for your feedback. We acknowledge your concern regarding the statement. We have revised the sentence to reflect a more balanced perspective.
2.8 - the objective and the gap are confusing and not very scientifically appealing.
2.8 – Response: Thank you for your valuable feedback. We have revised the paragraph to clarify the objective and the gap (lines: 87-106)
2.9 There is no adequate description of the phenomenon of falls in general (please refer to: https://www.who.int/news-room/fact-sheets/detail/falls) and in a hospital and residential environment, where elderly people are more prone to stay, and of the possible prevention strategies (please refer to: https://doi.org/10.1177/25160435241246344).
2.9 - Thank you for your valuable feedback. We have revised the introduction to include a more comprehensive description of the phenomenon of falls in general, as well as in hospital and residential environments, along with possible prevention strategies, in accordance with your suggestion (lines 90-96). We also added the two references 19 and 20.
2.10 In the section on materials and methods
What is meant by older people? (line 112) perhaps subjects over 60, as explained later? (line 131), please unify the concepts.
2.9 Response: Thank you for your feedback. We have added the definition of older persons (aged 60 years or more) in the first line of the Methods section (line:111).
2.10 The acronym UBS has never been explained before, but there is mentioned of BHU.
2.10 – Response: Thank you for your feedback. We have reviewed and ensured that all acronyms are properly defined, including the explanation of UBS as Basic Health Unit (BHU) (lines: 111; 133; 134; 136).
2.11 Paragraph 2.4 should be called measurement and not the outcome.
2.11 – Response: Done.
2.12 - The fall history, sociodemographic data, etc., should be data collected.
2.11 – Response: Done.
2.12 Selection bias was not considered; please indicate the limits. Please indicate the period of the study.
2.12 – Response: Thank you for your feedback. In line with the STROBE guidelines, we acknowledge that selection bias is a potential limitation of this study. The sample was drawn from older persons registered at a Basic Health Unit (BHU) in São Paulo, which may not be representative of the broader population of older adults. This selection may limit the generalizability of our findings to other settings or populations. Efforts were made to include a diverse range of participants; however, the inherent limitations of convenience sampling and the specific demographic characteristics of the BHU attendees could introduce selection bias. We have added in Study limitations “Another limitation is the participant group, which, although extensive, may not fully rep-resent the diversity of the Brazilian older population, particularly in different cultural and socioeconomic contexts. This introduces potential selection bias, as the sample was drawn from older persons registered at a BHU in São Paulo, Brazil and may not be repre-sentative of Brazilian population of older adults. Future studies should aim to include a more representative sample to enhance the external validity of the findings.” (lines: 312-318).
We have indicated the data collection period clearly in the manuscript. The data collection dates are specified in the first line below the "Data Collection, Inclusion, and Exclusion Criteria" section. "Data collection was conducted between February and August 2018 by a nurse during nursing consultations at the BHU, with an average duration of 40 minutes." (lines:185-186).
2.13. Please indicate the name of the statistician.
2.13. Response: Thank you for your feedback. We have indicated the name of the individual who performed the statistical analyses. (185-186).
2.14. In the discussion, how can you state what is stated in lines 245-248 without considering a control sample and selection bias?
2.14. Response: Thank you for your insightful feedback. We appreciate your concern regarding the statements made in lines 245-248 of the discussion. Given the absence of a control sample and the potential for selection bias, we have described our work as an association study and acknowledged its limitations. To address these limitations, we used robust statistical analyses to help support our findings. Specifically, after controlling for various sociodemographic variables, we identified significant associations between the fear of falling, history of falls, and depressive symptoms. At no point do we suggest causality; instead, we emphasize that our study highlights correlations.
Additionally, we have clearly outlined the study's limitations in the respective section, acknowledging the potential biases and the cross-sectional nature of our research.
We hope this explanation clarifies our approach and addresses your concern.
2.15. Please completely revise the discussion, conclusions, and abstract in light of the indications provided.
2.15 Response: Thank you for your detailed feedback and suggestions. We have revised the discussion, conclusions, and abstract sections in accordance with your recommendations.
Round 2
Reviewer 1 Report
Comments and Suggestions for Authors
Please include my full rfecomodations in article:
6.4. Response: Thank you for your feedback. We have provided a detailed explanation of the low Adjusted R² values, the methodology for the adjusted linear regression, and the complete linear models in our response for clarity. However, we believe that including this level of detail in the manuscript itself is unnecessary. The current descriptions in the paper provide sufficient information to identify the analytical pathway and understand the key findings.
Our models allow us to understand the independent effects of fear of falling and history of falls on depressive symptoms, while controlling for sociodemographic and clinical variables.
Explanation for Low Adjusted R²
The low Adjusted R² values indicate that while our models have statistically significant predictors, they explain only a small portion of the variance in depressive symptoms. This suggests that there are other factors, not included in our models, that also contribute to depressive symptoms among older adults. The complexity of depressive symptoms, which are influenced by a wide range of biological, psychological, and social factors, could account for the relatively low explanatory power of our models.
Methodology for Adjusted Linear Regression
The data were analyzed using the software Statistical Package for Social Sciences - SPSS 26 (SPSS Inc.). The analyses were conducted by the research coordinator, L.M.V. The descriptive analysis was presented in absolute and relative values (categorized variables) and measures of central tendency for continuous numerical variables of the GDS-15 and FES-I scales.
- Simple Linear Regression: Used to check the association between depressive symptoms (GDS-15), sociodemographic and health variables, fear of falling (FES-I), and fall history.
- Multiple Linear Regression: Depressive symptoms were adjusted for sociodemographic and health variables with p < 0.05: gender (p = 0.002), self-perception of health status (p = 0.009), and ADL (p = 0.036).
- Logistic Regression Models: Conducted using depressive symptoms (0 = without or 1 = with depressive symptoms) as the outcome variable and fear of falling or history of falling as independent variables. These models were adjusted for the same variables used in the multiple linear regression. A significance level of 5.0% was chosen for the tests, with 95.0% confidence.
Complete Linear Models
The complete linear models used in the study are as follows:
- Model 1: Association between FES-I and Depressive Symptoms (Unadjusted)
- Depressive Symptoms=β0+β1×FES-I + ϵ
- Model 2: Association between FES-I and Depressive Symptoms (Adjusted for Sociodemographic Variables)
- Depressive Symptoms=β0+β1×FES-I+β2×Gender+ϵ
- Model 3: Association between FES-I and Depressive Symptoms (Adjusted for Sociodemographic and Clinical Variables)
- Depressive Symptoms=β0+β1×FES-I+β2×Gender+β3×Self-Perception of Health Status+β4×ADL+ϵ
- Model 4: Association between History of Falling and Depressive Symptoms (Unadjusted)
- Depressive Symptoms=β0+β1×History of Falling+ϵ\
- Model 5: Association between History of Falling and Depressive Symptoms (Adjusted for Sociodemographic Variables)
- Depressive Symptoms=β0+β1×History of Falling+β2×Gender+ϵ
- Model 6: Association between History of Falling and Depressive Symptoms (Adjusted for Sociodemographic and Clinical Variables)
- Depressive Symptoms=β0+β1×History of Falling+β2×Gender+β3×Self-Perception of Health Status+β4×ADL+ϵ
We hope this detailed explanation addresses your queries.
6.5 -How have you modelled your logistic regression? Dependent and independent variables?
Response: Thank you for your query. The logistic regression models were developed with depressive symptoms (GDS ≥6: Yes=1; No=0) as the dependent variable. The independent variables included fear of falling (measured by the Falls Efficacy Scale-International) and history of falling. These models were adjusted for sociodemographic variables (gender) and clinical variables (self-perception of health status and basic activities of daily living)
Author Response
#Review 1.
Thank you very much for taking the time to review this manuscript. Please find the detailed responses below and the corresponding revisions highlighted in track changes in the re-submitted files.
- Comments and Suggestions for Authors
Please include my full recommendations in article:
- Response: Thank you for your insightful feedback. We have addressed your concerns in the revised manuscript as follows:
Low Adjusted R² in Table 4: The low Adjusted R² values indicate that, while our models have statistically significant predictors, they explain only a small portion of the variance in depressive symptoms. This suggests that other factors, not included in our models, also contribute to depressive symptoms among older adults. The complexity of depressive symptoms, influenced by a wide range of biological, psychological, and social factors, could account for the relatively low explanatory power of our models. We have added this explanation to the "Study Limitations and Strengths" section, highlighted in green.
Methodology for Adjusted Linear Regression: We have provided a detailed explanation of how the adjusted linear regression was conducted, including the steps involved and the variables considered. This information has been added to the "Statistical Analysis" subsection in the Methods section, with the revisions highlighted in green for your review.
Complete Linear Models: We have included the complete linear models used in our analysis, which detail the relationship between depressive symptoms and the independent variables (FES-I, history of falling) while controlling for sociodemographic and clinical factors. These models are now explicitly stated in the "Statistical Analysis" subsection.
We believe these additions provide the necessary clarity and detail. The specific revisions can be found highlighted in green in both the "Methods" and "Discussion" sections of the manuscript.
- How have you modelled your logistic regression? Dependent and independent variables?
- Response: Thank you for your inquiry regarding the logistic regression modeling. We have addressed your question by clarifying the methodology used for the logistic regression analysis.
We modeled depressive symptoms (coded as 0 = without depressive symptoms and 1 = with depressive symptoms) as the dependent variable. The independent variables included fear of falling (measured by the FES-I) and fall history, with adjustments made for the same covariates as in the multiple linear regression, such as gender, self-perception of health, and activities of daily living (ADL).
This addition has been highlighted in green in the "Statistical Analysis" section for your reference.
Reviewer 2 Report
Comments and Suggestions for Authors
The authors have completed all requested corrections in the abstract, methodology, discussion, and conclusion sections. The corrections have significantly enhanced the quality of the manuscript. I believe the revised version of the manuscript is suitable for publication in Healthcare.
Author Response
The authors have completed all requested corrections in the abstract, methodology, discussion, and conclusion sections. The corrections have significantly enhanced the quality of the manuscript. I believe the revised version of the manuscript is suitable for publication in Healthcare.
Response: Thank you very much for the comment. Thank you for the opportunity to improve our manuscript.
Reviewer 3 Report
Comments and Suggestions for Authors
Dear authors,
By addressing the fact that paragraphs 96-108 are duplicates of 109-121, the paper has been significantly improved and can now be considered for publication. I have only one final suggestion: you should provide a more detailed discussion on managing and preventing the risk of falling and ensure that future proposals are more coherent.
Please correct the appropriately referenced text [20] in the references section.
Kind regards
Author Response
“Dear authors,
By addressing the fact that paragraphs 96-108 are duplicates of 109-121, the paper has been significantly improved and can now be considered for publication.
Response: Thank you very much for the correction. The paragraph has been deleted.
I have only one final suggestion: you should provide a more detailed discussion on managing and preventing the risk of falling and ensure that future proposals are more coherent.
Response: Thank you very much for the opportunity to clarify. More details have been added (410-411, 420-421).
Please correct the appropriately referenced text [20] in the references section.
Response: The paragraphs in reference [20] were checked. The information complies with the reference.